# LncRNA LINC01537 Promotes Gastric Cancer Metastasis and Tumorigenesis by Stabilizing RIPK4 to Activate NF-κB Signaling

**DOI:** 10.3390/cancers14215237

**Published:** 2022-10-25

**Authors:** Guang-Yu Zhong, Jia-Nan Tan, Jing Huang, Sheng-Ning Zhou, Jin-Hao Yu, Lin Zhong, Dong Hou, Shi-Lin Zhi, Jin-Tao Zeng, Hong-Ming Li, Chu-Lian Zheng, Bin Yang, Fang-Hai Han

**Affiliations:** 1Department of Gastrointestinal Surgery, Sun Yat-sen Memorial Hospital, Sun Yat-sen University, Guangzhou 510275, China; 2Thyroid and Parathyroid Surgery Center, West China Hospital of Sichuan University, Chengdu 610041, China; 3Department of Colorectal Surgery, Guangdong Provincial Hospital of Chinese Medicine, Guangzhou 510120, China; 4Department of Operation Room, Sun Yat-sen Memorial Hospital, Sun Yat-sen University, Guangzhou 510275, China

**Keywords:** lncRNA, LINC01537, gastric cancer, RIPK4, TRIM25, NF-κB

## Abstract

**Simple Summary:**

At the time of diagnosis, many gastric cancer (GC) patients were often found to have reached an advanced stage, with a five-year overall survival rate of less than 30%. Therefore, a better understanding of the underlying mechanism of GC metastasis is essential for developing novel and effective treatments. The aim of this study was to assess the role of lncRNAs in GC metastasis. We confirmed that LINC01537 promoted the proliferation, invasion, and migration of GC cells in vitro, and promoted tumorigenesis and metastasis in vivo. Mechanistically, LINC01537 stabilizes RIPK4 by reducing the binding of RIPK4 to TRIM25, and reducing its ubiquitination degradation, thereby promoting the expression of the NF-κB signaling pathway.

**Abstract:**

Many studies reported that long noncoding RNAs (lncRNAs) play a critical role in gastric cancer (GC) metastasis and tumorigenesis. However, the underlying mechanisms of lncRNAs in GC remain unexplored to a great extent. LINC01537 expression level was detected using quantitative reverse transcription-polymerase chain reaction (qRT-PCR) and immunohistochemistry (IHC). Its biological roles in GC were then investigated using functional experiments. In order to investigate the underlying mechanism of LINC01537 in GC, RNA pull-down, RNA immunoprecipitation, and ubiquitination assays were performed. LINC01537 was significantly overexpressed in GC tissues and associated with a poor prognosis. Functional experimental results revealed that LINC01537 promoted the proliferation, invasion, and migration of GC cells. The animal experiments revealed that LINC01537 promoted tumorigenesis and metastasis in vivo. Mechanistically, LINC01537 stabilizes RIPK4 by reducing the binding of RIPK4 to TRIM25 and reducing its ubiquitination degradation, thereby promoting the expression of the NF-κB signaling pathway. According to our findings, the LINC01537-RIPK4-NF-κB axis promoted GC metastasis and tumorigenesis.

## 1. Introduction

Gastric cancer (GC) is a prevalent gastrointestinal tumor with a high morbidity and mortality rate [1,2]. Despite recent advances in diagnosis and treatment strategies [3], GC patients’ prognosis remains poor, especially in patients with distant metastases. At the time of diagnosis, many patients were often found to have reached an advanced stage, with a five-year overall survival rate of less than 30% [4]. Therefore, a better understanding of the underlying mechanism of GC metastasis is essential for developing novel and effective treatment and diagnostic strategies.

The lncRNAs are RNA transcripts with lengths of more than 200 nucleotides [5]. They are numerous and lack coding ability. Increasing evidence is showing that the abnormal lncRNA expression can readily contribute to the occurrence and metastasis of many cancers. In GC, the lncRNA LOC441461 was downregulated and promoted cell growth [6]. LINC02381 has been reported to inhibit the Wnt pathway, hence preventing the progression of GC [7]. Although the role of certain lncRNAs in GC has been revealed, there are still many lncRNAs that remain largely unknown.

Receptor interacting protein serine/threonine kinase 4 (RIPK4) is one of the receptor-interacting proteins (RIPs) that is abnormally expressed in several cancers, including bladder and cervical cancers [8,9,10]. However, the role of RIPK4 in GC remains unclear and requires further studies. RIPK4 has been reported to activate the NF-κB pathway, promoting the malignant phenotype of multiple cancers [11,12].

Ye et al. identified LINC01537 as one of nine hub lncRNAs that are associated with GC prognosis, by using the cancer genome atlas (TCGA) database [13]. Previous research has revealed the role of LINC01537 in lung cancer [14,15]. However, the mechanism by which LINC01537 promotes gastric cancer progression has not yet been elucidated. Our study demonstrated that a high expression level of LINC01537 is associated with a poor prognosis in the gastric cancer cohort in the TCGA database. Further studies have shown that LINC01537 is highly expressed in GC cell lines. Further in vivo and in vitro functional study results indicated that LINC01537 might promote GC’s proliferation, migration, invasion, and metastasis. The mechanistic studies found that LINC01537 stabilizes RIPK4 by reducing the binding of RIPK4 to TRIM25, and reducing its ubiquitination degradation, thereby enhancing the NF-κB signaling pathway to promote GC metastasis and tumorigenesis.

## 2. Materials and Methods

### 2.1. Cell Lines and Cell Culture

The human GC cell lines MGC-803, HGC-27, BGC-823, AGS, and normal gastric epithelial cell line GES-1, were obtained from the American Type Culture Collection. All cells were cultured in DMEM (Gibco, Carlsbad, CA, USA) that was supplemented with 10% fetal bovine serum and 1% penicillin-streptomycin (Invitrogen, Carlsbad, CA, USA). All cells were cultured at 37 °C in an incubator under a 5% CO_2_ environment.

### 2.2. Public Data Analysis

The cohort data for GC were retrieved from the Cancer Genome Atlas (TCGA) database, in order to explore whether a high expression level of LINC01537 is associated with a poor prognosis.

### 2.3. Clinical Specimens

Clinical samples were collected from Sun Yat-sen Memorial Hospital, Sun Yat-sen University (Guangzhou, China). After surgical resection, all tissues were immediately frozen in liquid nitrogen and stored at −80 °C. RT-PCR was used to verify LINC01537 expression levels in 59 pairs of freshly frozen specimens. Immunohistochemistry (IHC) was performed on 75 paraffin-embedded GC samples. The study was approved by the Medical Ethics Committee of Sun Yat-sen Memorial Hospital, Sun Yat-sen University, and written informed consent was obtained from all of the patients.

### 2.4. RNA Extraction and Real-Time PCR

RNA extraction: a volume of 1 mL of TRIzol reagent (Life Technologies, Carlsbad, CA, USA) and 200 μL of chloroform were added in turn, mixed by vortexing, and allowed to stand at room temperature for 5 min. It was then centrifuged at 12,000× *g* for 15 min at 4 °C, and the supernatant was taken. A volume of 400 μL of colorless liquid was drawn from the upper layer into a new enzyme-free EP tube, and the same amount of isopropanol was added, mixed upside down, and allowed to stand at room temperature for 10 min. The liquid was then centrifuged at 12,000× *g* for 10 min at 4 °C. The supernatant was poured out, and 1 mL of precooled 75% ethanol was added to each tube; the centrifuge tube was shaken upside down, and then centrifuged at 12,000× *g* for 5 min at 4 °C. the supernatant was discarded, and 20 μL of DEPC water was added in order to dissolve the precipitate after the RNA precipitate was dried. A Nanodrop2000 (Thermo, Waltham, MA, USA) instrument was used to detect the concentration and purity of RNA. The extracted RNA could be used for subsequent experiments or stored at −80 °C.

All RNA extractions were performed using a TRIzol reagent. Using a cDNA synthesis kit, the corresponding complement DNA (cDNA) was synthesized, following extraction (Bio-Rad, Hercules, CA, USA). A volume of 2 μL of 5× PrimeScript RT Master Mix per 500ngRNA was added, and the volume was replenished with DEPC water to 10 μL. The reaction conditions were as follows: 37 °C for 15 min, and 85 °C for 5 s. The cDNA products that were obtained by reverse transcription could be used for qPCR reactions.

PCRs were performed using an SYBR Green PCR kit, as per the manufacturer’s instructions. Each reaction well contained 5 μL of 2× ChamQ Universal SYBR qPCR Master Mix primer, 0.4 μL of forward and reverse primers (10 μm) of the target gene, 1 μL of cDNA. Water was added to fill the volume to 10 μL. All reagents were stored and added onto ice. After adding samples, the reagents were collected via brief centrifugation to the bottom. The level of RNA expression was calculated using the 2^−ΔΔct^ rule. GAPDH and MALTA were used as internal controls. The primer information used in our analysis is presented in Appendix A.

### 2.5. Western Blotting

Protein extraction: cells were discarded from the culture medium, washed twice with pre-cooled PBS, added with RIPA or IP lysis buffer containing protease inhibitors and phosphatase inhibitors (the volume of the lysis buffer was adjusted according to the bottom area of the culture vessel), and lysed on ice for 20 min. the cell lysate was transferred to a clean 1.5 mL EP tube with a cell scraper, centrifuged at 12,000× *g* for 20 min at 4 °C, and the supernatant was taken to a new EP tube.

Protein concentration measurement: solution A and solution B of BCA reagent (Beyotime, Shanghai, China) were combined in a ratio of 50:1, mixed evenly, with dilution of the protein standard in a gradient. Volumes of 10 μL of protein sample and 190 μL detection reagent were mixed, incubated at 37 °C for 30 min, and detected at a wavelength of 562 nm in a multi-plate reader for the absorbance value. A standard curve was drawn, according to the detection value of the gradient protein concentration standard, and the absorbance value of the detection sample was substituted to calculate the protein sample concentration. The required volume of lysate was calculated on the basis of the target loading volume. After adding 4× loading buffer at a ratio of 3:1, the sample was mixed gently, and placed in a constant temperature water bath at 98 °C for 10 min of denaturation. The obtained protein could be used in subsequent experiments, or stored at 20 °C for later use.

Western blotting: a BCA kit was used to the determine protein content (Beyotime, Shanghai, China). The exact amount of protein was separated using SDS-PAGE, and transferred to polyvinylidene fluoride. After 2 h of immobilization with 5% BSA, the corresponding primary antibodies were incubated overnight at 4 °C. The next day, after washing the band three times with TBST, the corresponding secondary antibodies were incubated. SyngenG: BOX Chemi XT4 Western Blotting was then used to detect the signals.

The antibodies were RIPK4 (ABclonal A8485, San Diego, CA, USA), RIPK4 (CST 12636s, Boston, MA, USA), NF-κB-P65 (Abcam ab16502, Cambridge, Cambridgeshire, UK), Ubiquitin (CST 3933s, Boston, MA, USA), IgG (ABclonal AC005, San Diego, CA, USA), and GAPDH (Proteintech 60004-1-Ig, Wuhan, China).

### 2.6. Transfection

For the overexpressions of LINC01537 and RIPK4, their sequences were cloned into the expression vector pcDNA3.1 (Invitrogen, Shanghai, China). The siRNAs of LINC01537 and RIPK4 were designed and synthesized by IGEbio (Guangzhou, China) to knock down LINC01537 and RIPK4. Small hairpin RNAs (shRNA) of LINC01537 were synthesized and cloned into the pLKO.1-EGFP-puro vector (IGEbio, Guangzhou, China). AGS or BGC-823 was seeded into 6-well plates at a density of 2.5–5 × 10^5^/well, gently shaken to evenly distribute the cells, and placed in an incubator overnight. SiRNA transfection was performed the next day, and the cell density reached 30% to 50%. Volumes of 5 μL of lipofectamine 3000 and 5 μL of siRNA/NC were diluted with 125 μL of Opti-MEM non-serum medium, respectively, stood at room temperature for 5 min, and mixed, continuing to incubate it at room temperature for 15–20 min. The original medium was discarded, and the cells were washed twice with PBS. A volume of 1.75 mL of Opti-MEM medium was added to each well, and the transfection mixture was added according to the pre-group, then gently mixed and put back into the incubator. After 6–8 h, the medium was replaced with ordinary complete medium, and after culturing for 48–72 h, the transfection efficiency was detected using qPCR and Western blot analysis. Subsequently, experiments were carried out. All siRNA sequences are listed in Appendix A. Complete medium: DMEM medium with 10% FBS (fetal bovine serum).

### 2.7. CCK-8 Assay and Colony Formation Assay

CCK8 was used to detect cell viability, as per the manufacturer’s protocol. Transfected or treated cells were resuspended using fresh complete medium, and cell counts were performed. A seeding rate of 2000 cells/200 μL was used in filling 96-well plates, with 3 replicate wells for each group. The CCK8 working solution was prepared according to the ratio of (CCK 10 μL + complete medium 100 μL)/well. The medium in the wells to be tested was discarded, and the prepared CCK8 working solution of 110 μL/well was added to the wells. The samples were incubated for 2 h in the incubator under darkness. A multifunction microplate reader (wavelength 450 nm) was used to detect the absorbance of the wells to be tested. A blank control well was set for each detection. The difference between the OD value of the well to be tested and the OD value of the blank well was the final value of each well. The experiment was repeated 3 times independently, and the data of each group were expressed as a mean value. When detecting cell proliferation, a total of 5 time points were detected on day 0, day 1, day 2, day 3, and day 4, and the cell proliferation curve was finally drawn.

In a six-well plate, 3 × 10^2^ cells were seeded per well, and cultured for two weeks for the colony formation assay. The medium was removed from the cells of each group, and the cells were washed 2–3 times with PBS. Paraformaldehyde at 4% concentration was fixed at room temperature for 15 min and then removed, and 0.5% crystal violet staining solution was added for staining at room temperature for 20 min. After the crystal violet solution was removed, the excess staining solution was carefully washed off, dried, and photographed to record the colony formation.

### 2.8. In Vitro Invasion and Migration Assay

For an invasion assay, 70 µL Matrigel was spread on the upper chamber. A total of 10^5^ cells were uniformly dropped into the upper chamber with serum-free DMEM, while the lower chamber contained a complete medium the next day. For the migration assay, 10^5^ cells were seeded in the top chamber without Matrigel. For these two assays, cells were incubated for 24 h, followed by removal of the cells on the upper surface of the chamber; the cells on the lower surface were fixed by 4% paraformaldehyde at room temperature for 15 min, stained by 0.5% crystal violet staining solution at room temperature for 20 min, and counted under a microscope.

### 2.9. Animal Experiment

The study was approved by Sun Yat-sen Memorial Hospital, Sun Yat-sen University. The mice were randomly divided into two groups. The first group of mice underwent subcutaneous tumorigenic experiments to observe the effect of LINC01537 on the tumorigenic potential of tumor cells; the second group was utilized to model liver metastasis. BGC cell lines (sh-LINC01537-NC (sh-NC), sh-LINC01537-01 (sh-1), and sh-LINC01537-02 (sh-2)) that stably knock down LINC01537, were successfully constructed using lentivirus vector. The subcutaneous tumorigenesis model or liver metastasis model was divided into 3 groups (sh-NC, sh-1 and sh-2), with 6 rats in each group. Briefly, five-week-old nude BALB/c mice were subcutaneously injected with 5 × 10^6^ sh-LINC01537 or sh-control cells in the flank in the subcutaneous tumorigenesis model. In the liver metastasis model, anesthetized with ether, the operation field skin was sterilized with 75% alcohol. A 1.0 cm oblique incision was made in the left upper abdomen of nude mice. A small amount of spleen was gently pulled out of the abdominal cavity with tissue tweezers. BGC cells (sh-NC, sh-1, or sh-2 BGC) were slowly injected into the spleens of nude mice, using a 1-milliliter syringe. Each nude mouse was injected with 0.1 mL of cell suspension (5 × 10^6^/ mouse), and the injection time was about 3–5 min. After injection in each group, the needle eye was compressed with 75% alcohol cotton for 3 min to stop bleeding and kill cancer cells that might leak out, or prevent intraperitoneal metastasis; the wound was then sutured. The operation process strictly complied with the principle of aseptic operation. After waking up from anesthesia, the SPF level was raised. The animals in the first group were sacrificed after six weeks, while the second group of animals was sacrificed after eight weeks, following injection. The tumors were dissected, and their size and weight were measured, analyzed, and compared. The tumor volume was calculated using the following equation:$$\mathrm{Tumor}\;\mathrm{volume}=\mathrm{Lenght}\;\times {\mathrm{Width}}^{2}/2$$

### 2.10. Wound Healing Assay

In a 6-well plate, 10^6^ cells were evenly distributed and cultured overnight. The next day, a scratch was made with a 200 µL pipette tip, followed by gentle washing with sterile PBS three times, in order to remove scratched cells. The scratches were then photographed at intervals of 0 and 72 h.

### 2.11. Nucleoplasmic Separation Assay

According to the instructions of the PARISTM protein kit and the RNA isolation system nucleoplasm separation kit, when the density of the cells reached 90% in a 15 cm dish, the medium was discarded and washed with pre-cooled PBS 2–3 times, centrifuged at 1000 rpm for 3 min, washed once with PBS, and centrifuged again; the cell pellets were then collected and placed on ice. The cells were then resuspended with 500 μL of pre-cooled cell fraction buffer in the kit, incubated on ice for 10 min, centrifuged at 500× *g* at 4 °C for 5 min, and then 200 μL of the supernatant was carefully aspirated as cytoplasmic component. The supernatant was completely removed, and the remaining pellet in the centrifuge tube was the nuclear component. RNA extraction of cytoplasmic components was continued according to the instructions of the nucleoplasm separation kit. Volumes of 1 mL of TRIzol and 200 μL of chloroform were added to the nuclear components, and RNA was extracted according to the previous steps.

### 2.12. RNA Pull-Down Assay and Truncation Assays

Biotin-labeled LINC01537 was synthesized in vitro according to the protocol. When preparing the cell lysate, the cell culture medium was first discarded, while the culture was washed twice with pre-cooled PBS; IP lysis buffer containing protease inhibitors and phosphatase inhibitors was added, and cells were lysed on ice for 30 min. Then, the lysate was collected and centrifuged at 12,000× *g* at 4 °C. The supernatant was collected after 30 min. The protein concentration of the supernatant should be greater than 1 mg/400 μL. Then, 1 mg of protein and 3 ug of biotin-labeled LINC01537/NC were mixed, supplemented with IPlysis containing protease inhibitors and phosphatase inhibitors to 500 μL, and RNA inhibitors were added. After incubation at room temperature for 2 h, biotin-affinity magnetic beads M280 (Thermo, Waltham, MA, USA) were added, and incubated at room temperature for 1 h to capture the RNA-protein complex. At the end of incubation, the magnetic beads were collected on a magnetic rack, and the supernatant was discarded. After being washed five times, the beads were dissolved in 1× LDS (Thermo, Waltham, MA, USA) and heated for 10 min at 98 °C. The precipitated proteins were then used for subsequent Western blot or mass spectrometry detection.

Truncation assays: the truncation experiment was carried out to clarify the binding sequence of LINC01537 to RIPK4. It was divided into 6 segments, according to each segment of 500 nt, namely, 0–2567 nt (F1), 0–500 nt (F2), 501–1000 nt (F3), 1001–1500 nt (F4), 1501–2000 nt (F5), and 2001–2567 nt (F6). Pull-down experiments (biotin-labeled sequences, F1–F6) were then performed to determine which sequences could bind to RIPK4.

### 2.13. RNA Immunoprecipitation Assay (RIP)

When the density of the cells reached 90% in a 15 cm dish, the medium was discarded and the cells were washed with pre-cooled PBS 2–3 times. The cells were lysed using 1 mL IP lysis with RNA inhibitors and protein inhibitors, at 4 °C for 20 min. Then, the cells were centrifuged at 12,000× *g* at 4 °C for 20 min, and the supernatant was collected. An amount of 5% of the volume as taken as input, and stored at −80 °C. The remaining supernatant was equally divided into two parts, and incubated with protein antibody RIPK4 antibody 5 μL, and IgG antibody 5 μL. The supernatants were incubated overnight at 4 °C with rotation, A/G magnetic beads (PIERCE PROTEIN A/G MAGNETIC, Thermo Fisher, Waltham, MA, USA) were added, followed by incubation with rotation at room temperature for 2 h. After incubation, the supernatant was washed twice with IP lysis. The A/G beads with the RNA/protein mixture were collected on the magnetic rack, and the supernatant was discarded. Quantities of 1 mL of TRIzol and 200 μL of chloroform were added to the A/G-RNA-protein mixture and the input, followed by mixing with vortexing, and allowed to stand at room temperature for 5 min. After centrifuging at 12,000× *g* for 15 min at 4 °C, the supernatant was taken, absolute ethanol was added to reach 2.5 times the volume; 1/10 volume of sodium acetate and 1 μL of glycogen were also added, and the solution was mixed well and stored at −80 °C overnight. Then, the mixture was centrifuged at 12,000× *g* for 20 min, until blue precipitation could be seen. The supernatant was discarded, and the precipitate was washed with precooled 75% and 95% ethanol, and centrifuged at 12,000× *g* for 5 min at 4 °C. The supernatant was discarded, and 20 μL of DEPC water was added to dissolve the precipitation after the RNA was dried. The LINC01537 expression level was then determined using qRT-PCR.

### 2.14. Ubiquitination Assay

The details of siRNA transfection can be seen in Section 2.5. Transfection. After 72 h of transfection, 2 mL complement medium with MG132 (10 µM/L) was added to each well for 24 h. The cells were then lysed using IP lysis containing protease inhibitors and phosphatase inhibitors (see protein extraction), and the same amount of lysate was incubated with protein antibody (RIPK4 antibody, 5 μL). After incubation overnight at 4 °C with rotation, A/G magnetic beads were added, followed by incubation with rotation at room temperature for 2 h. After incubation, the cells were washed five times with IP lysis. The A/G beads with RNA/protein mixture were collected on a magnetic rack, and the supernatant was discarded. After being washed five times with IP lysis, the beads were dissolved in about 30 μL of 1× LDS (Thermo), and heated for 10 min at 98 °C. The precipitated proteins were then used for subsequent Western blotting. The eluates were electrophoretically separated using SDS-PAGE, and incubated with an anti-ubiquitin antibody.

### 2.15. Immunohistochemical Scoring (IHC)

The staining intensity and the percentage of cells under the microscope were scored for each field as follows: no staining—0, light yellow—1, brown yellow—2, brown—3. The final IHC score was the product of the percentage of cells with microscopic staining intensity multiplied by the staining intensity score.

### 2.16. Statistical Analysis

All data were presented as means ± standard deviations (mean ± SD). The data were analyzed using SPSS and GraphPad Prism 7. Student’s *t*-tests and one-way ANOVAs were employed to analyze the statistical differences. Chi-square test was used to examine differences between variables. The survival difference was determined using Kaplan–Meier analysis and log-rank test. *p* < 0.05 was considered statistically significant.

## 3. Results

### 3.1. LINC01537 Is Highly Expressed in GC and Associated with a Poor Prognosis

According to the UCSC Genome Browser database, LINC01537 is located at chr11:72,281,700–72,284,266, and consists of 2567 bp (Appendix A). Online Coding Potential Assessment Tool (CPAT, https://lilab.research.bcm.edu/cpat/index.php, accessed on 1 June 2021) confirmed that LINC01537 did not have the ability to encode a protein (Appendix A). TCGA patients were classified as having high or low LINC01537 expression on the basis of cutoff value. The cutoff value was 0.04444, which was calculated using the survminor package in R (R X64 3.6.1). An expression level of LINC01537 that was higher than or equal to the cutoff value was defined as high expression of LINC01537 (LINC01537-high, *N* = 117), and an expression level that was lower than the cutoff value was defined as low expression of LINC01537 (LINC01537-low, *N* = 117). The Kaplan–Meier survival analysis demonstrated that GC patients with a high level of LINC01537 expression had lower overall survival (OS) and disease-free survival (DFS) (Figure 1A,B). In order to further validate it, qRT-PCR was performed on 59 GC tissues and compared to adjacent normal tissues, in order to confirm that LINC01537 was highly expressed in GC tissues (Figure 1C). According to our clinical data, Kaplan–Meier survival analysis revealed that GC patients with elevated LINC01537 expressions had lower OS and DFS (Figure 1D,E). We also analyzed the relationship between clinic-pathological factors and LINC01537 expression levels. LINC01537 expression level was related to N (Lymph node metastasis stage) and TNM stages (“TNM” stands for TNM Staging, T: tumor, N: node, M: metastasis.), as demonstrated in Table 1. The receiver operating characteristic (ROC) curve was used to evaluate the role of LINC01537 in predicting the TNM stage of GC. We defined stages IA, IB, IIA, and IIB as the “early stage” group, and stages IIIA, IIIB, and IIIC as the “late stage” group. LINC01537 expression levels in the “early stage” and “late stage” groups were then detected via qRT-PCR, and the results indicated that the expression of LINC01537 significantly increased in the “late stage” group (Figure 1F). The area under the curve (AUC) was found to be 0.669, suggesting that LINC01537 has a certain reference value for the differential diagnosis between the “early stage” group and the “late stage” group (Figure 1G).

### 3.2. LINC01537 Knockdown Inhibits Proliferation, Migration, and Invasion of GC Cells

We designed two siRNAs to investigate the role of LINC01537 in GC cells. The expression of LINC01537 in GC cell lines was detected using RT-PCR. The results showed that LINC01537 was highly expressed in AGS and BGC-823 (Figure 2A). Hence, these two cell lines were selected for siRNA transfection. qRT-PCR results revealed that the knockdown efficiency of these two siRNAs was approximately 70% (Appendix A). CCK8 assay revealed that knocking down the LINC01537 significantly inhibited the growth of AGS and BGC-823 (Figure 2A,B). The colony formation assay showed the same results as that of the CCK8 assay (Figure 2C,D). In vitro invasion and migration results indicated that silencing LINC01537 significantly inhibited invasion and migration (Figure 2E-H). A wound-healing assay also suggested that silencing LINC01537 reduced GC cell migratory ability (Figure 2I–L). After knockdown and overexpression of LINC01537 in normal gastric epithelial cell line GES-1, CCK8 assays and Transwell assays were conducted. We found that after knockdown of LINC01537, there was no significant change in the proliferation, invasion, and migration ability of GES-1. However, the overexpression of LINC01537 significantly enhanced the proliferation, invasion, and migration ability of GES-1 (Appendix A). In general, the knockdown of LINC01537 suppressed the malignant phenotype of GC cells.

### 3.3. LINC01537 Knockdown Inhibits Tumorigenesis and Metastasis of GC

In order to investigate the role of LINC01537 in vivo, the first group of mice underwent subcutaneous tumorigenic experiments so that the influence of LINC01537 on the tumorigenic ability of tumor cells could be observed. In contrast, the second group served as models for liver metastasis. BGC-823 that stably knocked down LINC01537 (sh-LINC01537-01 and sh-LINC01537-02) were successfully constructed for in vivo experiments. The volumes and weights of subcutaneous tumors in the sh-NC group were significantly larger than those in the sh-LINC01537-01 and sh-LINC01537-02 groups (Figure 3A–C). Expressions of LINC01537 in subcutaneous tumor and liver metastasis were detected via RT- PCR. The results showed that the expression of LINC01537 in the sh-NC group was significantly higher than that in the sh-LINC01537-01 and sh-LINC01537-02 groups (Figure 3D). We further explored the expression of RIPK4 and ki67 in subcutaneous tumors by IHC (See Section 2.14). Immunohistochemical scoring revealed that RIPK4 and Ki67 expression levels were significantly higher in the sh-NC group than in the sh-LINC01537-01 and sh-LINC01537-02 groups (Figure 3E–H). HE staining was then used to determine the number of liver metastases in different groups. In our liver metastasis model, the number of metastatic diseases in the sh-NC group was significantly higher than in sh-LINC01537-01 and sh-LINC01537-02 groups (Figure 3I,J), respectively.

### 3.4. LINC01537 Interacts with RIPK4 in GC Cells

In order to explore the possible mechanism of LINC01537, we first defined its intracellular localization. Nucleoplasmic separation assays revealed that LINC01537 was located in both the nucleus and cytoplasm (Figure 4A). In order to further explore the mechanism of LINC01537, we conducted a pull-down assay followed by silver staining, and mass spectrometry followed by identification of RIPK4 as the interactive protein of LINC01537 on the basis of its high score and Western blot results (Figure 4B,C). According to truncation assays, RIPK4 binds to the 1000–1500-nt segment of LINC01537 (Figure 4D). RIP assay revealed that RIPK4 antibody could enrich LINC01537 in AGS and BGC-823 (Figure 4E), implying that LINC01537 could bind to RIPK4.

### 3.5. LINC01537 Increases the Stability of RIPK4 by Reducing the Level of Ubiquitination

Previous research has suggested that RIPK4 expression may activate the NF-κB pathway, Wnt/β-catenin, RAF1/MEK/ERK, and STAT3 [12,16,17]. When LINC01537 was knocked down, the protein levels of β-catenin, RAF1/MEK/ERK, and STAT3 were detected via Western blotting. The results suggested that the levels of β-catenin, RAF1/MEK/ERK, and STAT3 did not change significantly, while the levels of NF-κB decreased significantly; hence, we focused on this pathway (Appendix A (source data can be found at Appendix A)). When LINC01537 was knocked down, qRT-PCR results revealed that the mRNA levels of RIPK4 remained unchanged (Appendix A). However, Western blotting indicated that the protein levels of RIPK4 and NF-κB were significantly reduced (Figure 4F). Therefore, we assumed that LINC01537 regulates RIPK4 via post-transcriptional regulation. Ubiquitination degradation is one of the most important mechanisms of post-transcriptional regulation. A reduction in RIPK4 protein level after siRNA transfection was found to be alleviated when MG132 was introduced to inhibit proteasome activity (Figure 4G). Cycloheximide (CHX) assays revealed that the half-life of the RIPK4 protein in sh-LINC01537 GC cell lines was shortened (Figure 4H). Ubiquitination assays suggested that knocking down LINC01537 enhanced the ubiquitination of RIPK4 in BGC-823 (Figure 4I). RIPK4 antibody was utilized to perform a pull-down assay, and mass spectrometry analysis of the proteins in NC and si-LINC01537 groups was used to further identify E3 enzymes that regulate RIPK4 ubiquitination (Figure 4J,K). The results showed that when LINC01537 was knocked out, the abundance of TRIM25 was 10-fold higher than in the NC group. Therefore, we conducted further experimental verification. LINC01537 knockdown did not significantly alter TRIM25 mRNA and protein levels (Appendix A (source data can be found at Appendix A)). The co-IP and WB experiments suggested that RIPK4 and TRIM25 may be mutually binding. The co-IP experiment revealed that the level of TRIM25 that was bound to RIPK4 significantly increased after LINC01537 was inhibited (Figure 4L,M). In conclusion, LINC01537 binds to RIPK4, and enhances its stability by lowering its ubiquitination level.

### 3.6. LINC01537 Promotes the Proliferation, Invasion, and Metastasis of GC Cells by Activating RIPK4-NF-κB Pathway

In order to determine whether LINC01537 mediated GC cell growth, invasion and migration depends on the RIPK4-NF-κB pathway; ectopic RIPK4-overexpression plasmid or empty control was transfected into si-NC or si-1 GC cells. The expression level of RIPK4 in cells was detected using qRT-PCR (Figure 5A) and Western blot analysis (Appendix A (source data can be found at Appendix A)). The expression level of RIPK4 in cells was detected with qPCR. The protein level of NF-KB was detected using Western blotting. The results suggested that the protein level of NF-KB decreased after knockdown of LINC01537. After replenishing RIPK4, the decreased protein level of NF-KB was completely relieved (Appendix A (source data can be found at Appendix A)). CCK8 assay, colony formation assay, Transwell assay, and wound healing assay results revealed that RIPK4 overexpression could reverse the inhibitory effect of LINC01537 knockdown on proliferation, invasion, and metastasis (Figure 5B–L).

We next wanted to clarify whether the overexpression of LINC01537 could increase the malignant progression of GC cells, and whether LINC01537 mainly acts through the NF-κB pathway. NF-κB inhibitor JSH-23 or empty control was administrated into NC or LINC01537-overexpression GC cells. The expression level of NF-κB in cells was detected using Western blot analysis (Figure 6A). CCK8 assay (Figure 6B), colony formation assay (Figure 6C,D), wound healing assay (Figure 6E–H), and Transwell assay (Figure 6I–L) results revealed that overexpression of LINC01537 can promote the proliferation, invasion and migration of GC cells, while the administration of JSH-23 could dramatically inhibit LINC01537-induced malignant progression. By activating the RIPK4-NF-κB pathway, LINC01537 promotes the malignant phenotype of GC cells.

### 3.7. RIPK4 and NF-κB-p65 Expressions in Clinical Specimens

In order to confirm that LINC01537-mediated invasion augmentation is dependent on the RIPK4-NF-κB pathway in vivo, immunohistochemical (IHC) experiments were performed on tissue sections to detect the expression levels of RIPK4 and NF-κB-p65; moreover, correlations between the expression levels of LINC01537, RIPK4, and NF-κB-p65 proteins in clinical samples were further analyzed. The IHC results show the positive or negative of RIPK4 and NF-κB-p65 (Figure 7A,B). According to Spearman correlation analysis, the LINC01537 expression level was significantly and positively correlated with RIPK4 and NF-κB-p65 expression levels (Figure 7C,D). We defined IHC scores that are greater than 1 to be positive, and less than or equal to 1 as negative (RIPK4—negative, N = 43; RIPK4—positive, N = 32; NF-κB—negative, N = 45; NF-κB—negative, N = 30). The results of Kaplan–Meier survival analysis showed that RIPK4 and NF-κB-p65 protein expressions are significantly related to the prognosis of GC patients. GC patients with high RIPK4 expression have a worse OS and DFS (*p* < 0.05) (Figure 7E,F). GC patients with high NF-κB-p65 expression had a worse prognosis (*p* < 0.05) (Figure 7G,H). Next, we investigated the relationship between the RIPK4 protein expression levels in clinical specimens, and the basic clinical-pathological characteristics. Table 2 displays the results. The RIPK4 protein expression level was significantly correlated with the N and TNM stages of GC patients (*p* < 0.05). In contrast, there was no correlation with the age, gender, T stage, and differentiation degree of GC patients (*p* > 0.05), suggesting that RIPK4 protein expression in GC tissue has a certain predictive effect on the lymph node metastasis and TNM staging of GC patients. In conclusion, we verified that LINC01537 expression is positively correlated with RIPK4/NF-κB-p65 expression, and that GC patients with high RIPK4/NF-κB-p65 have a worse prognosis, proving that LINC01537 promotes the malignant phenotype of GC cells by activating the RIPK4/NF-κB-p65 signaling pathway.

We constructed a graphical illustration of LINC01537 activating NF-κB-p65 by stabilizing RIPK4 and reducing its ubiquitination degradation (https://www.figdraw.com/static/index.html#/, on 9 July 2022) (Figure 8).

## 4. Discussion

In recent years, increasing evidence has shown that lncRNAs play key roles in cancer development and metastasis [18,19]. Nine lncRNAs that were associated with prognoses from another article on the identification of ceRNA network related to gastric carcinogenesis and prognosis, and were reported in Ye’s article. Among them, four have been explored for their functions and mechanisms in gastric cancer; one of them has the ability to encode proteins; two of them have not yet identified their full-length sequences; and one of them is not related to both OS and DFS. All of the above are excluded, thus our research in this article focused on LINC01537. Previous research revealed that LINC01537 may inhibit the occurrence and development of lung cancer. However, its involvement in GC is unclear. Therefore, the role of LINC01537 in GC was further studied in this study. This study identified lncRNA LINC01537 to be highly expressed in GC tissue, while lowly expressed in adjacent normal tissue, something which was also identified in clinical tissue samples with qRT-PCR. According to TCGA data, GC with high LINC01537 expression had worse OS and DFS. The gain and loss functional experiment revealed that LINC01537 could promote the proliferation, invasion, and migration of GC cells in vitro and in vivo. This indicates that LINC01537 plays a key role in GC metastasis and development.

Further investigation of its mechanism showed that LINC01537 performs a cancer-promoting role, mainly by directly binding with RIPK4, and promoting the activation of the NF-κB pathway. The RIPK4 protein is one of the members of the RIP kinase family, and is a serine/threonine kinase whose important role in activating the NF-κB pathway has been demonstrated [20]. Huan Yi et al. suggested that RIPK4 can promote EMT transformation and ovarian cancer metastasis [21]. According to QI et al., high expression of RIPK4 promotes pancreatic cancer metastasis [16]. However, its specific mechanism in GC remains unknown. In our investigation, knocking down LINC01537 lowered the protein levels of RIPK4 and NF-κB-p65, but did not affect the mRNA level of RIPK4. We believe that regulation of LINC01537 on RIPK4 is post-transcriptional. Therefore, we further explored how LINC01537 regulates RIPK4 protein levels. Ubiquitination is an important mechanism of post-transcriptional regulation, and we predicted the existence of ubiquitination sites in RIPK4 on a prediction website [22,23]. The results suggest that RIPK4 has a ubiquitination site. Our assumption was validated using MG132, CHX, and ubiquitination assays. The results of CHX assays revealed that knocking out LINC01537 significantly reduced the half-life of RIPK4 protein, demonstrating that LINC01537 may protect the RIPK4 protein against degradation and proteolysis. The decrease in the protein level of RIPK4 after siRNA transfection was alleviated when MG132 was added to inhibit proteolysis. Further studies suggested that TRIM25 is an E3 ligase that regulates RIPK4 ubiquitination. The tripartite motif protein family is characterized by a ring finger domain that functions as an E3 ligase [24]. Some members of the TRIM family are involved in the development and progression of the tumor, including autophagy, cell proliferation, metastasis, and apoptosis [25,26]. Our findings imply that after LINC01537 is knocked down, RIPK4 binds to TRIM25 and promotes RIPK4 ubiquitination. Therefore, it may be concluded that LINC01537 promotes the malignant phenotype of GC cells via regulating the ubiquitination level of RIPK4 protein.

NF-κB is an essential transcription factor that plays a critical role in the occurrence and metastasis of various cancers [27]. By inducing the transcription of multiple key genes, the activation of the NF-κB signal can regulate important biological processes, such as proliferation, epithelial-mesenchymal transition, metastasis, and angiogenesis [28,29,30]. Maubach G. et al. discovered that Helicobacter pylori induces gastric carcinogenesis by activating NF-κB [31]. The role of the RIPK4-NF-κB axis in promoting the invasiveness of urothelial bladder carcinoma, and a promising therapy of delivery of RIPK4 small interfering RNA for bladder cancer, was reported by Jian Ye Liu et al. [11,32]. However, there is no relevant information on the role of the RIPK4-NF-κB axis in GC. Our functional experimental results indicated that NF-κB protein levels decreased significantly after knocking down LINC01537, but were restored following transfection with exogenous RIPK4. In conclusion, LINC01537 binds directly to RIPK4, reducing its ubiquitination level and leading to an increase in RIPK4 levels in the cell. Abrogation of RIPK4 degradation promotes the accumulation of RIPK4 in GC cells, and activates the NF-κB-p65 pathway.

This study concluded that LINC01537 activates the NF-κB pathway in GC cells by stabilizing the RIPK4 protein. The role of the non-coding RNA LINC01537-RIPK4-NF-κB pathway in GC proliferation and metastasis was established. Our study provides more in-depth evidence for advancing the mechanistic study of non-coding RNAs in GC metastasis.

## 5. Conclusions

The elevated expression of LINC01537 in GC tissue is indicative of a poor prognosis. LINC01537 enhances the proliferation, invasion, and migration of GC cells by binding directly to RIPK4, thereby reducing its ubiquitination level and consequently activating the NF-κB signaling pathway.

## Figures and Tables

**Figure 1 cancers-14-05237-f001:**
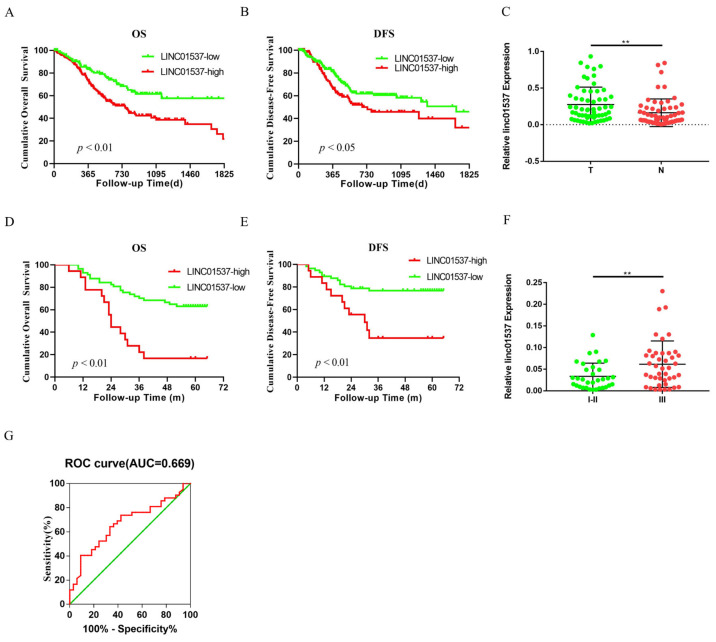
High expression of LINC01537 is associated with poor diagnosis in GC. OS (**A**) and DFS (**B**) were compared between LINC01537 high and low expressions in GC patients from the TCGA cohort (Kaplan-Meier survival analysis, *N* = 334). (**C**) Expression levels were compared between 59 GC and adjacent normal tissues (Student’s *t*-test, *N* = 59). OS (**D**) and DFS (**E**) were compared between LINC01537 high and low expressions in GC patients in our hospital (Kaplan-Meier survival analysis, *N* = 75; LINC01537-high, *N* = 18; LINC01537-low, *N* = 57). (**F**) Expression levels of LINC01537 were compared between stage I–II and stage III patients. (**G**) ROC curves were used to evaluate the role of LINC01537 in predicting the TNM stage of GC. The survival difference was determined using Kaplan-Meier analysis and log-rank test (**A,B**,**D**,**E**). Student’s *t*-tests (**C**,**F**) were employed to analyze the statistical differences. ** *p* < 0.01, m: month.

**Figure 2 cancers-14-05237-f002:**
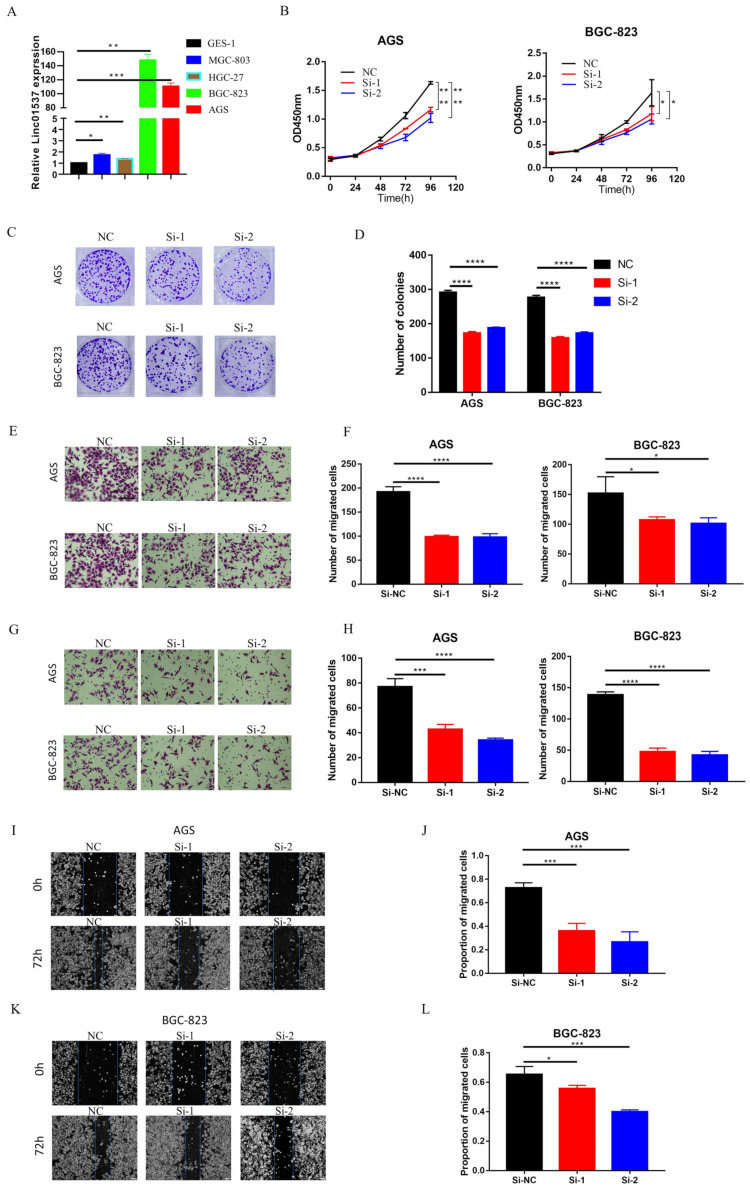
LINC01537 knockdown inhibits proliferation, migration, and invasion of GC cells. (**A**) Expression levels of LINC01537 in GES-1, MGC-803, HGC-27, BGC-823, and AGS were detected via qRT-PCR. After being transfected with siRNA (NC: negative control; Si-1: Si-LINC01537-01; Si-2: Si-LINC01537-02), CCK8 assays were used to detect the effect of LINC01537 expression on AGS and BGC-823 cell proliferation (**B**). Colony assays was conducted to test for tumorigenesis (**C**,**D**). Transwell assays were used to detect the effect of LINC01537 expression on migration (**E**,**F**) and invasion (**G**,**H**) abilities. Wound healing was used to detect the effect of LINC01537 expression on cell migration in AGS (**I**,**J**) and BGC-823 cells (**K**,**L**). Scale bar (seen in the first figure of Figure 2E) is 100 μm. Figure 2E,G were taken under 20X objective lens. Figure 2I,K were taken under 10X objective lens. Student’s *t*-tests (**A**) and one-way ANOVAs (**B**–**K**) were employed to analyze the statistical differences. * *p* < 0.05, ** *p* < 0.01, *** *p* < 0.001, **** *p* < 0.0001.

**Figure 3 cancers-14-05237-f003:**
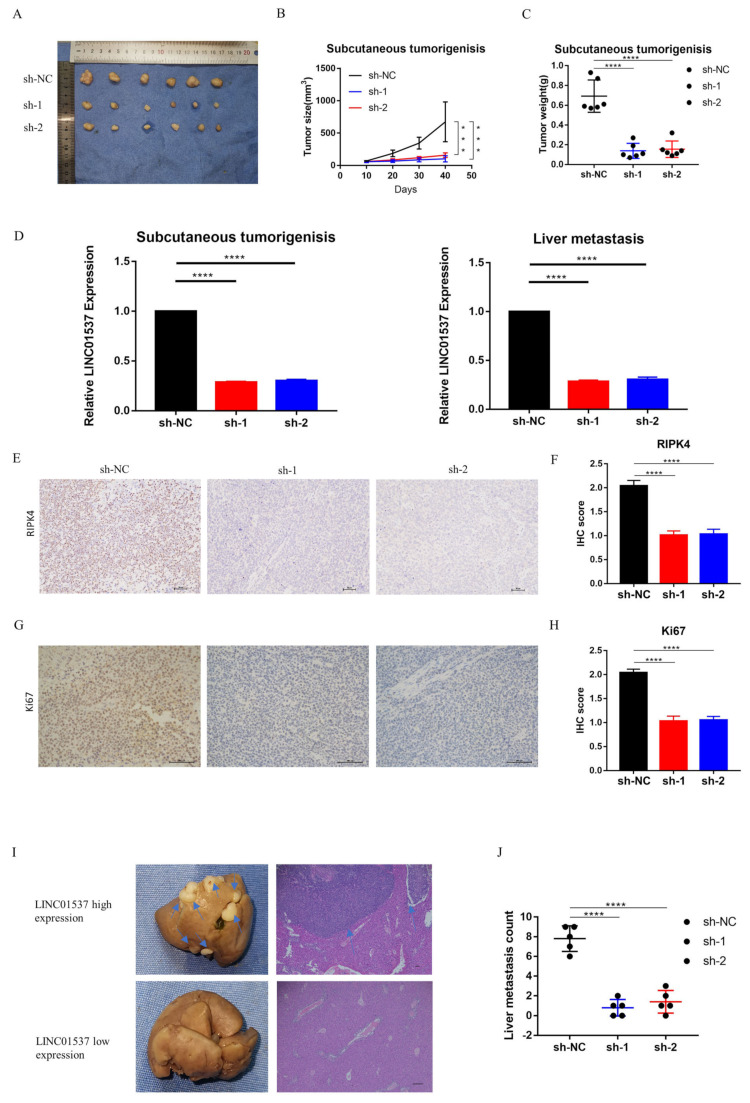
LINC01537 promotes the tumor growth and metastasis of GC in vivo. BGC cells transfected with shRNA (NC: negative control; sh-1: sh-LINC01537-01; sh-2: sh-LINC01537-02) were injected into the flank (results shown in **A**–**H**) or spleen (results shown in **I**,**J**) of nude BALB/c mice, in order to detect the effect of LINC01537 expression on tumorigenesis and metastasis in vivo. (**A**) The subcutaneous tumors in different groups at indicative time points are shown. (**B**,**C**) The tumor size and tumor weight of subcutaneous tumors in the sh-NC group were compared with those in sh-LINC01537-01 and sh-LINC01537-02 groups. (**D**) qRT-PCR was used to detect the expression levels of LINC01537 in subcutaneous tumors and metastatic nodules. (**E–H**) IHC analysis of the expressions of RIPK4 and Ki67, and their corresponding IHC scores. (**I**) Hematoxylin and eosin (HE) staining of liver tissues were used to detect liver metastasis nodules. (**J**) The metastatic nodules in the livers of nude mice were counted. Figure 3E,G were taken under 10X objective lens. Figure 3I (right panels) were taken under 4X objective lens. All data are expressed in the form of mean ± SD. One-way ANOVA was employed to analyze the statistical differences. *** *p* < 0.001, **** *p* < 0.0001.

**Figure 4 cancers-14-05237-f004:**
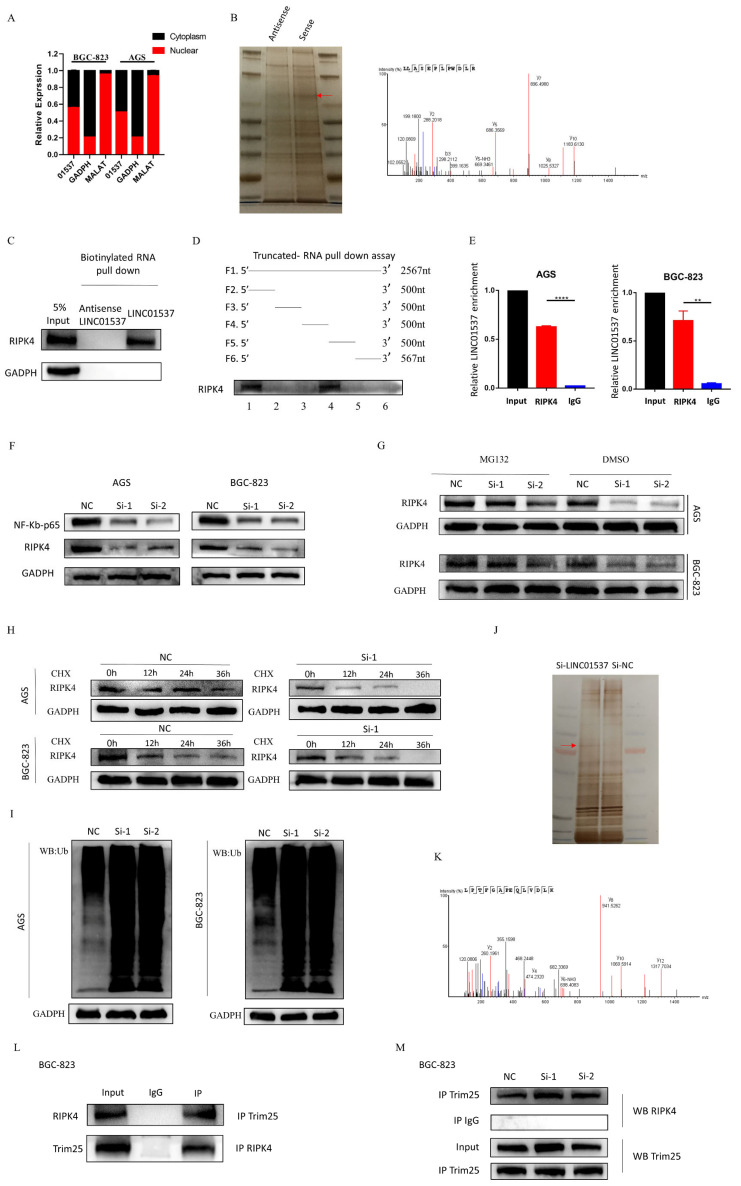
LINC01537 binds to RIPK4 and increases the stability of RIPK4 by reducing the level of ubiquitination. (**A**) Nucleoplasmic separation assays were conducted to verify whether LINC01537 was located in the nucleus or cytoplasm (GADPH served as cytoplasmic reference, while MALAT served as nuclear reference). (**B**) Identification of potential interactive protein of LINC01537 via RNA pull-down assay followed by silver staining, and mass spectrometry. (**C**) The possible interactive protein of LINC01537 was identified via Western blot analysis. (**D**) Truncation assays were conducted to identify the segment of LINC01537 that RIPK4 binds to. (**E**) RIP assays were used to verify RIPK4 and LINC01537 combinations. (**F**) Western blot analysis of RIPK4 and NF-κB-p65 in control, and LINC01537-knockdown AGS and BGC-823. (**G**) Western blot analysis of RIPK4 in control, and LINC01537-knockdown AGS and BGC-823 treated with MG132 or DMSO. (**H**) Western blotting of RIPK4 in control, and LINC01537-knockdown AGS and BGC-823 treated with CHX at the indicated timepoints. (**I**) Knockdown of LINC01537 enhances the ubiquitination levels of RIPK4 in AGS and BGC-823. (**J**,**K**) Identification of potential interactive E3 enzyme of RIPK4 in control, and LINC01537-knockdown AGS and BGC-823 by silver staining. The possible interactive E3 enzyme of RIPK4 was identified via Western blotting. (**L**) Endogenous interaction between RIPK4 and TRIM25 was determined using co-immunoprecipitation with anti-RIPK4 or anti-TRIM25 antibodies in BGC-823 cells. (**M**) Endogenous interaction between RIPK4 and TRIM25 was determined using co-immunoprecipitation with anti-RIPK4 or anti-TRIM25 antibodies in si-NC or si-1 BGC-823 cells. The uncropped blots are shown in supplematary materials, named as “Source Western-blot Images for Figure XX”. NC: negative control; Si-1: Si-LINC01537-01; Si-2: Si-LINC01537-02. CHX: cycloheximide. Student’s *t*-tests were employed to analyze the statistical differences. ** *p* < 0.01, **** *p* < 0.0001 (source data can be found at Appendix A).

**Figure 5 cancers-14-05237-f005:**
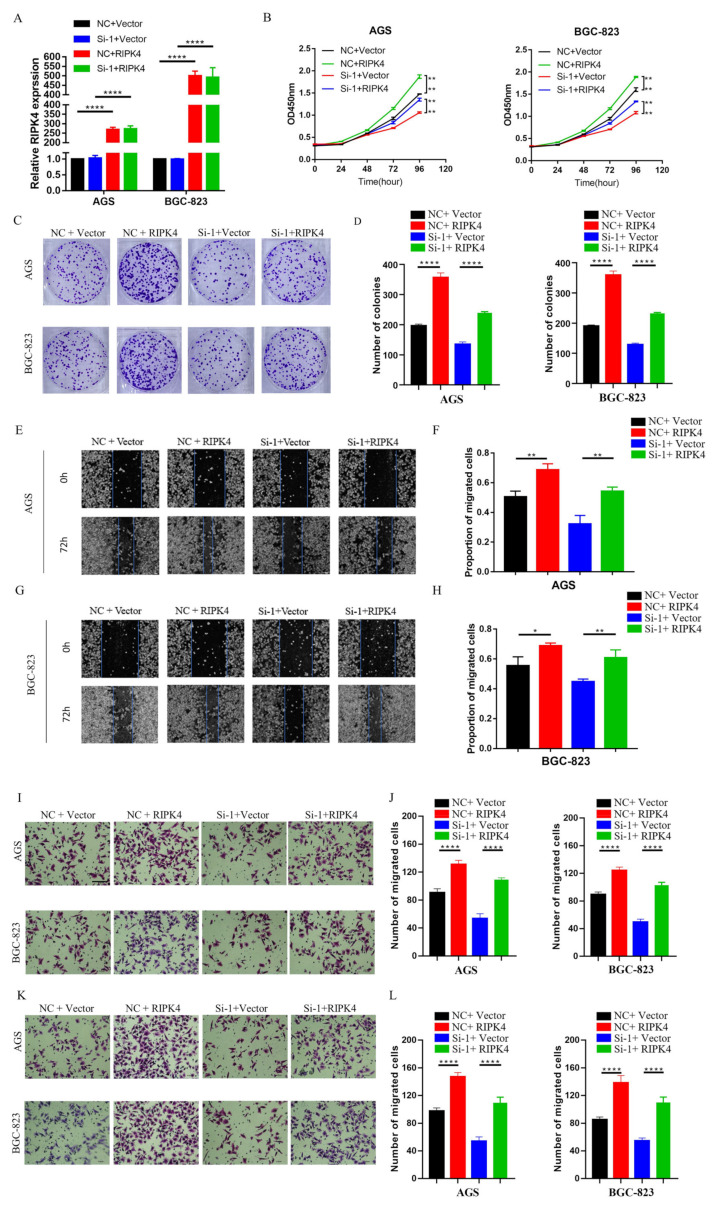
LINC01537 promotes proliferation, migration, and invasion of GC cells by activating the RIPK4-NF-κB pathway. qRT-PCR (**A**) and Western blot (**B**) analysis of the levels of RIPK4 in si-NC or si-1 BGC-823, and AGS transfected with ectopic RIPK4-overexpression plasmid, or empty control. CCK8 assays (**B**) and colony assays (**C**,**D**) were used to detect ectopic expression of RIPK4, which rescued the decreased proliferation and tumorigenesis ability induced by LINC01537 knockdown. Whether ectopic expression of RIPK4 could rescue the decreased migration and invasion ability induced by LINC01537 knockdown are shown by wound healing assay (**E**–**H**) and Transwell assays (migration ability shown in (**I**,**J**), invasion ability shown in (K,**L**)). Figure 5E,G were taken under 10X objective lens. Figure 5I,K were taken under 20X objective lens.NC + Vector: negative control + empty vector; NC + RIPK4: negative control + ectopic RIPK4; Si-1 + vector: Si-LINC01537-01 + empty vector; Si-1+ RIPK4: Si-LINC01537-01 + ectopic RIPK4. One-way ANOVA was employed to analyze the statistical differences. * *p* < 0.05; ** *p* < 0.01; **** *p* < 0.0001.

**Figure 6 cancers-14-05237-f006:**
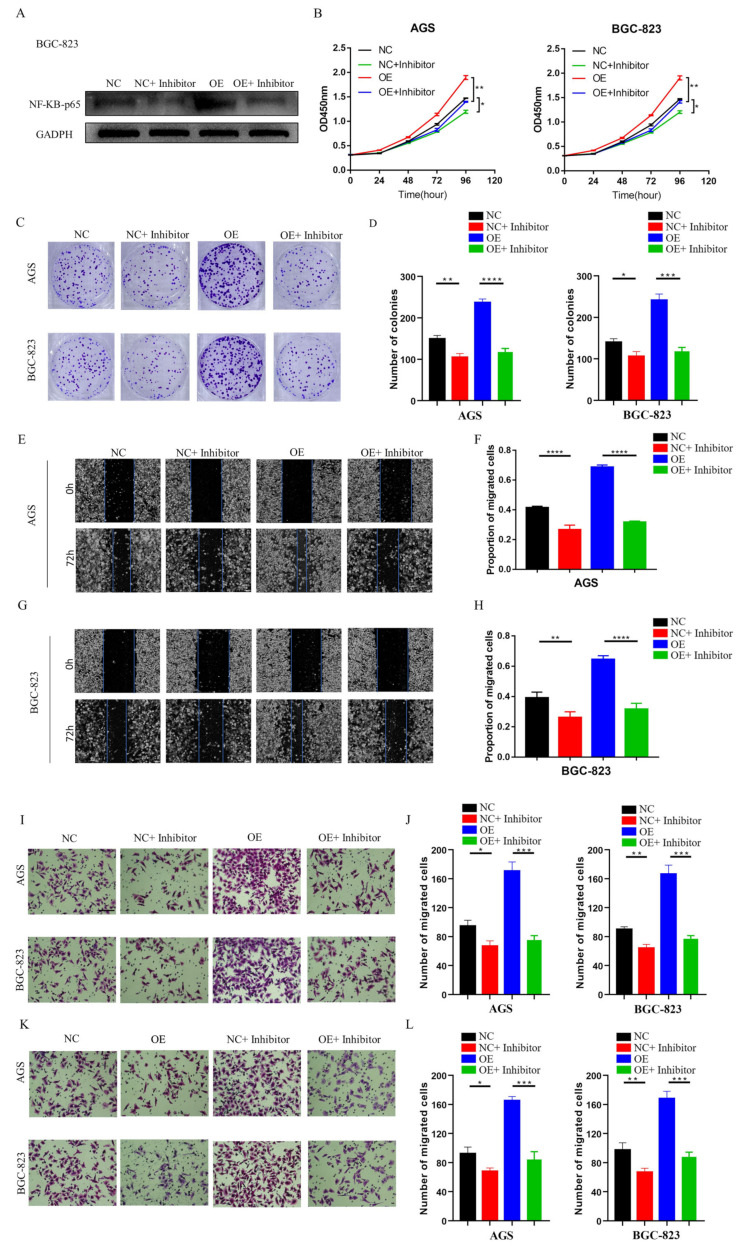
Blocking the NF-κB pathway reverses LINC01537-induced GC progression in vitro. Western blot (**A**) analysis of the protein levels of NF-κB in NC or OE (LINC01537-overexpression BGC-823 administrated with NF-κB inhibitor or not) (source data can be found at Appendix A). CCK8 assays (**B**) and colony assays (**C**,**D**) were used to detect whether administration of NF-κB inhibitor could inhibit the increased proliferation and tumorigenesis ability induced by LINC01537 overexpression. Wound healing assay (**E**–**H**) and Transwell assays (migration ability shown in (**I**,**J**), invasion ability shown in (**K**,**L**)) were used to detect whether administration of NF-κB inhibitor could inhibit the increased migration and invasion ability induced by LINC01537 overexpression. Figure 6E,G were taken under 10X objective lens. Figure 6I,K were taken under 20X objective lens. NC: negative control; NC: negative control + NF-κB inhibitor; OE: overexpression of LINC01537; OE + inhibitor: overexpression of LINC01537 + NF- κB inhibitor. One-way ANOVA was employed to analyze the statistical differences. * *p* < 0.05; ** *p* < 0.01; *** *p* < 0.001; **** *p* < 0.0001.

**Figure 7 cancers-14-05237-f007:**
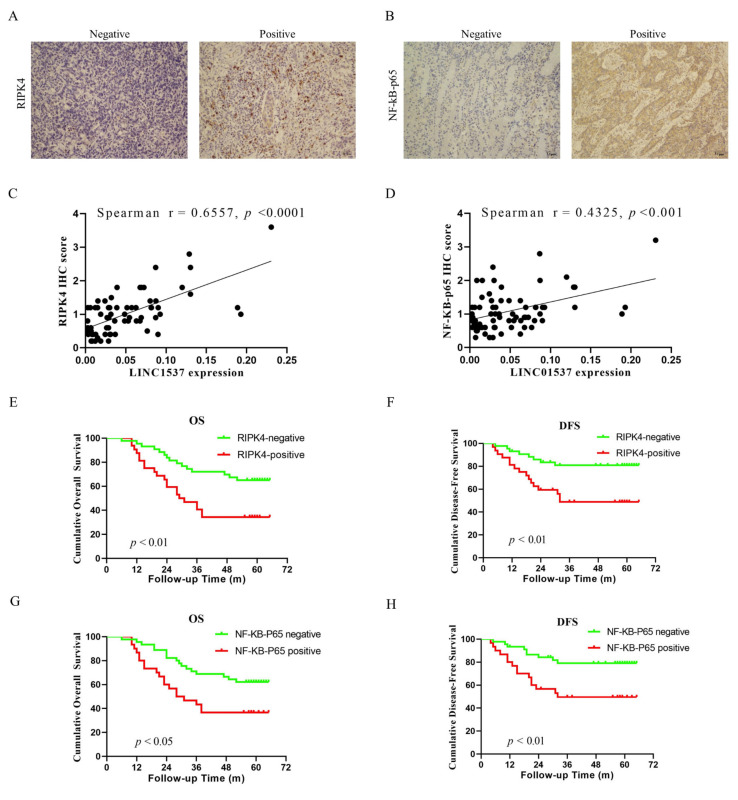
Clinical specimens prove that the high expressions of RIPK4 and NF-κB -p65 in GC tissue were associated with poor prognoses. (**A**,**B**) The expression levels of RIPK4 and NF-κB -p65 in IHC clinical specimens. (**C**) Correlation analysis of expression levels of LINC01537 and RIPK4 in clinical specimens. (**D**) Correlation analysis of expression levels of LINC01537 and NF-κB -p65 in clinical specimens. OS (**E**) and DFS (**F**) were compared between RIPK4 high and low expressions in GC patients (RIPK4—negative, N = 43; RIPK4—positive, N = 32). OS (**G**) and DFS (**H**) were compared between NF-κB -p65 high and low expressions in GC patients (NF-κB—negative, N = 45; NF-κB—positive, N = 30). Spearman correlation analysis was used to analyze correlations between LINC01537 expression and RIPK4 and NF-κB -p65 IHC scores (**C**,**D**). The survival difference was determined using the Kaplan–Meier analysis and log-rank test (**E**–**H**).

**Figure 8 cancers-14-05237-f008:**
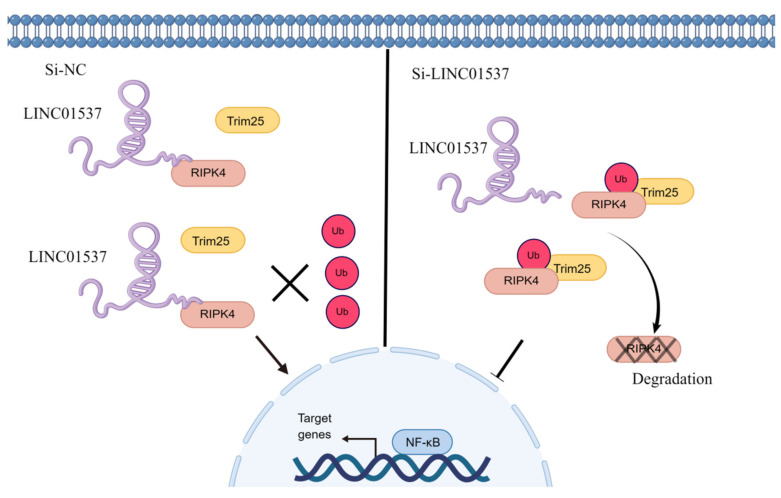
A graphical illustration of LINC01537 activating NF-κB-p65 by stabilizing RIPK4 and reducing its ubiquitination degradation.

**Table 1 cancers-14-05237-t001:** The clinic-pathological factors of GC patients.

Clinic-Pathological Factors	LINC01537 Expression		*p*-Value
	Low Expression (*N* = 57) (%)	High Expression (*N* = 18) (%)	
Age	55.75 ± 11.93	56.61 ± 10.39	0.7853
Gender			0.2488
Male	41 (71.93)	10 (55.56)	
Female	16 (28.07)	8 (44.44)	
N stage			**0.0218**
N0	23 (40.35)	2 (11.11)	
N ≥ 1	34 (59.65)	16 (88.89)	
T stage			0.4855
T1/T2	14 (24.56)	3 (16.67)	
T3/T4	43 (75.44)	15 (83.33)	
TNM stage			**0.0074**
I-II	30 (52.63)	3 (16.67)	
III	27 (47.37)	15 (83.33)	
Differentiation degree			0.0788
Well/moderately differentiated	14 (24.56)	1 (5.56)	
Poorly differentiated	43 (75.44)	17 (94.44)	

Data presented as absolute numbers of patients (%), or as mean ± standard deviation. Student’s *t*-tests were employed to analyze the statistical differences. The chi-square test was used to examine differences between variables. *p* < 0.05 was considered as statistically significant.

**Table 2 cancers-14-05237-t002:** The clinic-pathological factors of GC patients based on the expressions of RIPK4 and NF-κB-p65.

Variable	RIPK4		*p*–Value	NF-κB-p65		*p*–Value
	Negative (N = 43) (%)	Positive (N = 32) (%)		Negative (N = 45) (%)	Positive (N = 30) (%)	
Age	57.42 ± 11.69	54.00 ± 11.16	0.2058	57.00 ± 12.55	53.63 ± 10.17	0.2090
Gender			0.2134			0.6141
Male	32 (74.42)	19 (59.38)		32 (71.11)	19 (63.33)	
Female	11 (25.58)	13 (40.62)		13 (28.89)	11 (36.67)	
N stage			**0.0151**			0.3228
N0	20 (46.51)	6 (18.75)		18 (40.00)	8 (26.67)	
N ≥ 1	23 (53.49)	26 (81.25)		27 (60.00)	22 (73.33)	
T stage			0.5826			0.5780
T1/T2	11 (25.58)	6 (18.75)		9 (20.00)	8 (26.67)	
T3/T4	32 (74.42)	26 (81.25)		36 (80.00)	22 (73.33)	
TNM stage			**0.0203**			0.1581
I–II	24 (55.81)	9 (28.13)		23 (51.11)	10 (33.33)	
III	19 (44.19)	23 (71.87)		22 (48.89)	20 (66.67)	
Differentiation degree			0.5618			0.0771
Well/moderately differentiated	10 (23.26)	5 (15.63)		12 (26.67)	3 (10.00)	
Poorly differentiated	33 (76.74)	27 (84.37)		33 (73.33)	27 (90.00)	

Data presented as absolute number of patients (%), or as mean ± standard deviation. Student’s *t*-tests were employed to analyze the statistical differences. The chi-square test was used to examine differences between variables. *p* < 0.05 was considered as statistically significant.

## Data Availability

Data is contained within the article or supplementary material.

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
