# Peer review of "LncRNA LINC01537 Promotes Gastric Cancer Metastasis and Tumorigenesis by Stabilizing RIPK4 to Activate NF-κB Signaling"

_cancers, 2022, doi:10.3390/cancers14215237_

Round 1

Reviewer 1 Report (Previous Reviewer 1)

The authors updated the manuscript. But the authors should provide a clean version for checking. Even though some more information is included, the manuscript (including the supplementary) still needs serious revision for further consideration. The following comments are some examples, please check all the others by yourself.

1.      Please consistent the formation throughout the manuscript.

2. Please include more details materials information for other researchers to repeat your work relatively easier.

3.      Please include statistical methods in all the figure legends.

4.     The liver metastasis model should be in detail.

5.   The volume should be “μl” instead of “ul”, check throughout the manuscript.

6.   It should be the “student’s t-test”. Please correct this throughout the manuscript.

7.     Please use high-resolution figures.

8. Line 392: Italic “p” please, check throughout the manuscript including figure images.

9.  Line 437 Figure 1: Please horizon Figure 1C and 1F “X-axis” legend. Extend “T” and “N”, and check throughout the manuscript.

10. Line 452: How about knockdown of LINC01537 impacts normal cells?

11. Line 473 Figure 2C and similar others: Are the six images or upper/lower panel three images from the same 6-well plate? If yes, I’d highly suggest the authors use the “plate image”.

12. Line 515 Figure 3: Which group does the linc01537 high/low sample come from? How did you define high or low?

13. The NF-κB signaling pathway has a broad function. How do you know this is the major signaling pathway that mediates LINC01537?

Author Response

Thank you very much for your comments on our article. We will clarify the issues you raised in the comments one by one.

  1. Please consistent the formation throughout the manuscript.

Reply 1:The formation throughout the manuscript has been unified.

  1. Please include more details materials information for other researchers to repeat your work relatively easier.

Reply 2:More details materials information has been supplemented.

  1. Please include statistical methods in all the figure legends.

Reply 3:The statistical methods in all the figure legends throughout the manuscript has been supplemented.

  1. The liver metastasis model should be in detail.

Reply 4:Anesthetized with ether, the operation field skin was sterilized with 75% alcohol. A 1.0cm oblique incision was made in the left upper abdomen of nude mice. A small amount of spleen was gently pulled out of the abdominal cavity with tissue tweezers. BGC cells(sh-NC, sh-1 or sh-2 BGC ) were slowly injected into the spleen of nude mice with the a 1ml syringe. Each nude mouse was injected with 0.1ml of cell suspension (5×106/ mouse), and the injection time was about 3-5 minutes. After injection in each group, the needle eye was compressed with 75% alcohol cotton for 3 minutes to stop bleeding and kill cancer cells that might leak out, prevent intraperitoneal metastasis, and suture the wound. The operation process should strictly comply with the principle of aseptic operation. After waking up from anesthesia, the SPF level was raised.

  1.  The volume should be “μl” instead of “ul”, check throughout the manuscript.

Reply 5:The "ul" in the article has been replaced by "μl".

  1. It should be the “student’s t-test”. Please correct this throughout the manuscript.

Reply 6:The "student t-test" in the article has been replaced by "student’s t-test".

  1. Please use high-resolution figures.

Reply 7:High-resolution figures were supplemented.

  1. Line 392: Italic “p” please, check throughout the manuscript including figure images.

Reply 8:I checked throughout the manuscript carefully and changed all P values to italic P.(Line 392 and Figure 7)

  1. Line 437 Figure 1: Please horizon Figure 1C and 1F “X-axis” legend. Extend “T” and “N”, and check throughout the manuscript.

Reply 9:I checked throughout the manuscript carefully and horizon “X-axis” legend.

  1. Line 452: How about knockdown of LINC01537 impacts normal cells?

Reply 10: Corresponding supplementary experiments in the normal gastric epithelial cell line GES-1 were conducted. After knockdown and overexpression of LINC01537 in normal gastric epithelial cell line GES-1, CCK8 assays and Transwell assays were conducted. We found that after knockdown of LINC01537, there was no significant change in the proliferation, invasion and migration ability of GES-1. However, the overexpression of LINC01537 significantly enhanced the proliferation, invasion and migration ability of GES-1 (seen in Fig. S3)

  1. Line 473 Figure 2C and similar others: Are the six images or upper/lower panel three images from the same 6-well plate? If yes, I’d highly suggest the authors use the “plate image”.

Reply 11:The six images or upper/lower panel three images were not from the same 6-well plate. If necessary, we can repeat the experiments in the same 6-well plate.

  1. Line 515 Figure 3: Which group does the linc01537 high/low sample come from? How did you define high or low?

Reply 12:Linc01537 high samples was from sh-NC group while Linc01537 low was from sh-1 and sh-2 group. High or low is based on the level of LINC01537 expression. The sh-1 and sh-2 groups, BGC-823 that stably knocked down Linc01537 (sh-LINC01537-01 and sh-LINC01537-02) while in sh-NC group, BGC-823 without knocked down Linc01537 were injected into the flank or spleen.

  1. The NF-κB signaling pathway has a broad function. How do you know this is the major signaling pathway that mediates LINC01537?

Reply 13: We applied NF-kB inhibitor and conducted functional experiments. The functional experiments showed that the proliferation, invasion and migration functions of gastric cancer cells were significantly enhanced after overexpression of LINC01537. With the application of an inhibitor of NF-kB, this functional result was significantly inhibited (seen Figure7 for details). We validated several common pathways activated by RIPK4 reported in other literature. The results suggested that the level of β-catenin did not change significantly after knocking down LINC01537.

Reviewer 2 Report (Previous Reviewer 2)

The authors have improved the manuscript and answered all my comments.  Reviewers are working voluntarily to help improve the author's work and manuscript. Authors should show gratitude to all reviewers in the response letter. 

Author Response

   Thank you very much for your valuable comments. We have benefited a lot from it and have revised the article according to your comments.

Round 2

Reviewer 1 Report (Previous Reviewer 1)

Thank you for the updates. Please consider the following concerns, especially the statistical information that needs to be clear. Please provide a clean version and be consistent with the format throughout the manuscript. Thank you.

1) In most cases, a space is needed before the citation. Check throughout the manuscript.

2) Please be consistent with “min” or “minutes” throughout the manuscript.

3) Line 205: It should be “105”. The “5” should be superscript. Check throughout the manuscript.

4) Line 224: A space is needed before “FBS”.

5) Line 499: There is an extra space before “PCR”.

6) Line 523 Figure 2: a) How did you use the student’s t-test to analyze line curves in Figure 2B? b) Most of the experiments were designed with multiple groups. Why did the authors use the student’s t-test to analyze the data instead of the proper one, one-way ANOVA? c) line 534: It should be “μ”.

7) Line 567 Figure 3, line 700 Figure 5, line 718 Figure 6: a) How did you use the student’s t-test to analyze line curves in Figure 3/5/6 B? b) Most of the experiments were designed with multiple groups. Why did the authors use the student’s t-test to analyze the data instead of the proper one, one-way ANOVA?

Author Response

Thank you very much for your careful review and correction of our statistical methods. We will clarify the issues you raised in the comments one by one.

1)In most cases, a space is needed before the citation. Check throughout the manuscript.

Reply 1:A space has been added before the citation  throughout the manuscript.

2)Please be consistent with “min” or “minutes” throughout the manuscript.

Reply 2:"min" has been consistent throughout the manuscript.

3)Line 205: It should be “105”. The “5” should be superscript. Check throughout the manuscript.

Reply 3: I have check throughout the manuscript and found two mistakes, then corrected them. The “5” in line 205 and “6” in line 293 have been superscript.

4)Line 224: A space is needed before “FBS”.

Reply 4: A space has been added before “FBS”.

5)Line 499: There is an extra space before “PCR”.

Reply 5: I tried to delete it, but failed. This may be because it is the first word of a sentence.

6)Line 523 Figure 2: a) How did you use the student’s t-test to analyze line curves in Figure 2B? b) Most of the experiments were designed with multiple groups. Why did the authors use the student’s t-test to analyze the data instead of the proper one, one-way ANOVA? c) line 534: It should be “μ”.

Reply 6: a-b) Thank you very much for pointing out our statistical errors. At the same time, we also consulted statistical analysis experts, and used one way anova for statistics of all experiments designed by multiple groups. The results are shown in Figure 2,3,5,6 and Figure S. c) I have correct it.

7) Line 567 Figure 3, line 700 Figure 5, line 718 Figure 6: a) How did you use the student’s t-test to analyze line curves in Figure 3/5/6 B? b) Most of the experiments were designed with multiple groups. Why did the authors use the student’s t-test to analyze the data instead of the proper one, one-way ANOVA?

Reply 7: a-b) Thank you very much for pointing out our statistical errors. At the same time, we also consulted statistical analysis experts, and used one way anova for statistics of all experiments designed by multiple groups. The results are shown in Figure 2,3,5,6 and Figure S.

Round 3

Reviewer 1 Report (Previous Reviewer 1)

Dear Authors, thank you for the update. Please seriously deal with your manuscript to be consistent with the format. The followings are some examples, please seriously check all the others by yourself.

1: Please don't just list the methods at the end of figure legends except if only one method has been used. For example, in Figure 2, please list one-way ANOVA or student's t-test by the end of each panel. Check all the others.

2: Please be consistent with "h" or "hours"

3: Please be consistent with or without a space before or after the sign, "<" etc.

4: In Table 1 legend, please italic p.

5: Line 755: Please italic p.

Author Response

Thank you very much for your careful review and correction. We will clarify the issues you raised in the comments one by one.

1: Please don't just list the methods at the end of figure legends except if only one method has been used. For example, in Figure 2, please list one-way ANOVA or student's t-test by the end of each panel. Check all the others.

Reply 1:  The use of different statistical methods in different panels has been supplemented at the end of figure legend of (Figures 1, 2 and 7).

2: Please be consistent with "h" or "hours"

Reply 2: "h" has been consistent throughout the manuscript.

3: Please be consistent with or without a space before or after the sign, "<" etc.

Reply 3:A space has been added before and after the sign “<” throughout the manuscript.

4: In Table 1 legend, please italic p.

Reply 4:It has been changed to italic P.

5: Line 755: Please italic p.

Reply 5:It has been changed to italic P.

This manuscript is a resubmission of an earlier submission. The following is a list of the peer review reports and author responses from that submission.

Round 1

Reviewer 1 Report

The manuscript titled “LncRNA LINC01537 promotes gastric cancer metastasis and tumorigenesis by stabilizing RIPK4 to activate NF-κB signaling” describes the investigation of Linc01537 contribute to gastric carcinogenesis via RIPK4- NF-κB pathway. Overall, the materials and methods, figure legends, and related descriptions lack details that need seriously revise. The followings are some concerns and comments have been pointed out that the authors may want to consider.

1) Line 92: Materials and Methods section: a) Please include detailed reagents information, cat#, and so on. b) Please detailed methods to make your work relatively easier to repeat.

2) Line 211: Please use italic p as it refers to a p-value throughout the manuscript.

3) Lines 222-223: Please specify high or low expression and cutoff value with details.

4) Line 234: Please define “N” and “TNM”.

5) Lines 235-239: Please detail Figure 1G analysis/meaning in the methods section or descript it in the results section to make it clear.

6) Line 241 Figure 1: a) Please include statistical methods in the figure legend. b) Please include the sample size. c) Please italic p and be consistent with or without a space before and after the signs, eg. “<”. d) It’s hard to notice Figure 1D with a statistical difference. e) Please define “*” and “m”. f) The authors have already known the patients’ outcomes from the database, how can you prognosis?

7) Line 250 Table 1: a) Please include the total number. b) Please include the statistical method. c) Please reorganize most left column to make it clearer to read.

8) Line 272 Figure 2: a) Please update the figure legend with detailed information. It’s hard to read and understand. b) Please include scale bars.

9) Line 300 Figure 3: Please update the figure legend with detailed information. It’s hard to understand.

10) Line 352 Figure 4: a) Please update the figure legend with some more necessary detailed information. For example, statistical methods, and so on. b) Please provide higher resolution Figure 4B and 4K.

11) Line 389 Figure 5: a) Please update the figure legend with detailed information. It’s hard to read and understand. b) Please include scale bars. c) Why do the wounds seem with different sizes at 0h?

12) Line 435 Figure 6: Please update the figure legend with some more necessary detailed information. For example, statistical methods, and so on.

13) Line 446 Figure 7: Please include software or online tools that support general the figure.

14) Line 449 Table 2: a) Please adjust the table to make it looks good. There are lots of half brackets. b) Please include the total sample size. c) Please include statistical methods, and so on.

15) Line 484: Please confirm all capital letters for the family name.

16) Table S1: Please include the directions.

17) Figure S1 is difficult to read the details, please provide higher resolution images if possible and include the direct website link for easier tracking.

18) Figure S2, S3, S4: a) Please include sample size. b) Please include statistical methods.

19) Please well-labeled original images for easier tracking or nobody knows what those are.

20) More information is needed after the conclusion. For example, acknowledgment, and so on.

Reviewer 2 Report

Please see the attached word file. 
